# Plastic Entanglement Poses a Potential Hazard to European Hedgehogs *Erinaceus europaeus* in Great Britain

**DOI:** 10.3390/ani13152448

**Published:** 2023-07-28

**Authors:** Emily Thrift, Pierre Nouvellet, Fiona Mathews

**Affiliations:** School of Life Sciences, University of Sussex, Brighton BN1 9RH, UK; ed343@sussex.ac.uk (E.T.); pierre.nouvellet@sussex.ac.uk (P.N.)

**Keywords:** wildlife, rehabilitation centres, plastic waste, population modelling

## Abstract

**Simple Summary:**

Plastic entanglement is well known for causing both conservation and welfare issues for marine mammals, but little is known about the impacts on terrestrial species. Following anecdotal reports in the media, we assessed the prevalence and consequences of plastic entanglement for the European hedgehog (*Erinaceus europaeus*) in Great Britain. Based on data provided by rescue centres and population modelling, we estimate that 4000–7000 hedgehog deaths occur annually occur as a consequence of plastic entanglement, representing a significant welfare issue and placing additional pressure on a declining species.

**Abstract:**

A questionnaire to gather evidence on the plastic entanglement of the European hedgehog (*Erinaceus europaeus*) was sent to 160 wildlife rehabilitation centres in Great Britain. Fifty-four responses were received, and 184 individual admissions owing to plastic entanglement were reported. Death was the outcome for 46% (*n* = 86) of these cases. A high proportion of Britain’s hedgehogs enter rehabilitation centres annually (approximately 5% of the national population and potentially 10% of the urban population), providing a robust basis for assessing the minimum impacts at a national level. We estimate that 4000–7000 hedgehog deaths per year are attributable to plastic, with the true rate likely being higher, since many entangled hedgehogs—in contrast to those involved in road traffic accidents—will not be found. Population modelling indicates that this excess mortality is sufficient to cause population declines. Although the scale of the impact is much lower than that attributable to traffic, it is nevertheless an additional pressure on a species that is already in decline and presents a significant welfare issue to a large number of individuals.

## 1. Introduction

Plastic production has increased significantly since the 1950s, resulting in a global accumulation of plastic waste, which totalled 379.3 megatons (MT) in 2021 alone [1]. There is now concern about the ecological impacts of this waste [2]. The size of the plastic waste is important when considering what risks it poses; for example, macroplastics (defined as pieces of plastic >10 mm) [3] pose entanglement and gut blockage risks, whereas other risks may be presented by mesoplastics (size range 1 ≤ 10 mm) and microplastics (MPs; <1 mm) [3]. There are also concerns about the leaching of plastic additives and plasticizers, such as bisphenol A and phthalates, which are potential endocrine-disrupting chemicals (EDCs), refs. [4,5,6,7] and the adsorption and accumulation of toxins, including EDCs and heavy metals [8,9,10,11] on plastic particles. Recently, there has been substantial research on the impacts of all these types of plastic waste within marine habitats [12,13,14,15,16,17,18,19,20,21,22,23,24,25,26,27]. A recent review indicated, for example, that almost 40% of the 123 known marine mammals have been reported as becoming entangled in mesoplastics [28]. A further study by Butterworth and colleagues reports that entanglement has caused widespread suffering and death among marine mammals and birds [17]. Our understanding of the scale and impact of plastic entanglement on terrestrial species is, by comparison, very limited. Most studies on this topic have focused on birds [29,30,31,32], though entanglement is also reported in mammals such as the black howler monkey (*Alouatta caraya*), white-tailed opossum (*Didelphis albiventris*), fat mice (*Thylamys* sp.), polar bear (*Ursus maritimus*), and artic fox (*Alopex lagopus*) [19,33,34]. A study of a range of mammals and birds in Argentina found that 60% of the reported entanglement cases resulted in death due to asphyxiation or starvation [33]. A further study of both agricultural and urban crows indicated that 100% of nestlings that become entangled (*n* = 11) were unable to fledge and, in most cases, had long-term injuries, mainly to their toes [29]. The injuries and deaths reported in these terrestrial studies indicate that plastic entanglement potentially poses a risk to terrestrial wildlife. However, the scale of these risks relative to other factors, and the types of plastic involved, are poorly understood.

In Great Britain, there is growing concern about the impacts of plastic entanglement on the European hedgehog (*Erinaceus europaeus*). Plastic poses risks to both populations and individuals. Potentially even small increases in fatality rates could be important at a national level in a species classed as vulnerable to extinction [35], and entanglement also compromises individual welfare. This study, therefore, seeks to quantify the rate and severity of plastic entanglement of the European hedgehog and to estimate the likely impacts at the population level.

## 2. Materials and Methods

### 2.1. Questionnaire

An eight-question survey was created using Qualtrics software v. 2021 (Qualtrics, Provo, UT, USA) to gather data on the number of annual admissions, plastic-related admissions, and outcomes at British wildlife rehabilitation centres (Table A1). The survey was sent via email to 160 centres identified from a British database (directory.helpwildlife.co.uk accessed on 15 July 2019) and Google searches using the terms ‘hedgehog’ AND ‘hospital’ OR ‘rescue’. The survey covered the period from August 2019 to August 2020 and from October 2021 to October 2022, with the latter period targeting centres that had not previously completed the questionnaire to increase the response rate. The centres were asked to report on any cases they had seen in the previous 12 months. Incidents were recorded as entanglement regardless of whether it was the primary or secondary reason for admission. Furthermore, if an individual was known to have been released from plastic prior to admission, this was also recorded as an entanglement case.

### 2.2. Statistical Analyses

Analyses were carried out in R Studio base package [36]. Chi-square tests were used to assess the relationships between survival outcome and the predictor variables habitat type and plastic type. Wilson’s 95% confidence intervals were calculated using the Wilson.ci function.

### 2.3. Habitat Type

The locations at which hedgehogs were reported were mapped using the Geographical Information System Package QGIS (QGIS 3.28.3, 2019. The rescue centres (*n* = 52) were asked to provide the location of the site at which the hedgehog was reported as entangled; however, in some cases, this was not possible, and the site of the rescue centre was used (*n* = 5 centres). These habitats were then classified into three types (urban, peri-urban, and rural) using Google Earth (Google LLC, Google Earth version 7.3 2023). The peri-urban locations were classified as areas with less than 30% built cover.

### 2.4. Mortality Model

A population dynamics model was developed to assess the likely impacts of plastic entanglement-associated fatalities on population stability. We describe the dynamics of the hedgehog population as a (Leslie) matrix population:$$N_{t}=\left(\begin{array}{c} n_{1,t}\\ n_{2,t}\\ n_{3,t} \end{array}\right)$$
with $n_{i,t}$ representing the number of hedgehogs in age class ‘*i*’, i.e., 1 for individuals aged between 1 day to 15 weeks, 1 and 2 for those less than 2 years old, and 3 for adult and sexually active individuals. We implicitly assume an age class 0 for individuals 0–15 weeks old, which is not observed. The population number refers to the sizes 15 weeks after the birth pulse.

We define the mortality for each age class ‘*i*’ as $\mu_{i}$; female fecundity of sexually mature females as f, and assume a 1:1 sex ratio. Therefore, we have:$$N_{t+1}=M\;N_{t}$$
with $M=\left[\begin{array}{ccc} 0 & \frac{f\left(1-\mu_{0}\right)}{2}\left(1-\mu_{2}\right) & \frac{f\left(1-\mu_{0}\right)}{2}\left(1-\mu_{3}\right)\\ \left(1-\mu_{1}\right) & 0 & 0\\ 0 & \left(1-\mu_{2}\right) & \left(1-\mu_{3}\right) \end{array}\right]$.

The dominant eigenvalue, λ, of *M* gives us the population growth rate (i.e., $\lambda =N_{t+1}/N_{t}$), while the associated eigenvector, $\nu =\left(\begin{array}{c} \nu_{1}\\ \nu_{2}\\ \nu_{3} \end{array}\right)$, gives us the stable age distribution (e.g., the proportion of individuals in each age class).

Hedgehog population dynamics in Great Britain are uncertain, but some key estimated parameters relevant to us are highlighted in Table 1.

Using the population dynamics model, the combinations of estimated parameters from Table 1, and the 2 estimated growth rates, we estimate the overall mortality of adults’ hedgehogs (i.e., $\mu_{2}=\mu_{3}$), and the stable age distribution (ν) by numerically solving the equation M ν=λ ν.

Then, we aim to attribute all deaths ($D_{T}$) to 3 main causes: road traffic accidents (RTA) ($D_{R}$), plastic entanglement ($D_{P}$), and other causes ($D_{O}$).

It has recently been reported that 50% of hedgehogs in rescue centres are released [28], implying that 50% die, so the number of deaths in our sample ($D_{t}^{rescue})\;$ was 6273. This all-cause fatality rate is similar to that reported by our respondents (53.2% total mortality). Rescue centres provided estimates of plastic entanglement and other causes of death for 12,546 individual admissions. This revealed that plastic entanglement occurred in 184 hogs, from which $D_{P}^{rescue}=85$ died. For RTAs, we took estimated figures from 3 studies, owing to the absence of information from our dataset. This provides a low, medium and high range.

Assuming these figures reflect deaths in a natural environment, we estimate that among hedgehogs not dying from RTA, a proportion

$p_{O}^{rescue}=\frac{D_{O}^{rescue}}{D_{O}^{rescue}+D_{P}^{rescue}}$ die from other causes.

Taken altogether, we have:$$D_{P}=D_{T}-D_{R}-D_{O}$$

with
The total number of deaths: $D_{T}=N\;\sum \left(\left[\begin{array}{ccc} 0 & 0 & 0\\ \left(1-\mu_{1}\right) & 0 & 0\\ 0 & \left(1-\mu_{2}\right) & \left(1-\mu_{3}\right) \end{array}\right]\nu \right)$, with N The total population size and $\mu_{1}$ from Table 2, and $\mu_{2}$ and ν (age class stable distribution, i.e., eigenvector) estimated above;The total number of RTA ($D_{R}$) informed by the literature (Table 2);The total number of deaths linked to other causes: $D_{O}=\left(D_{T}-D_{R}\right)\;p_{O}^{rescue}$.

## 3. Results

Of the 160 rescue centres contacted, responses were received from 52 (Figure 1). These centres provided data on 12,546 hedgehog admissions. Ten centres reported zero admissions of plastic-entangled hedgehogs, and all of these reported annual admissions of fewer than 200 individuals; however, there were also centres with similar admission rates which did reported entanglement incidents. Data from the 44 centres with entanglement cases showed 184 admissions (1.4% of their total admissions) were a consequence of plastic entanglement.

More cases were recorded in urban than in other habitat types (Figure 2A). However, habitat was not associated with survivorship (χ^2^ = 1.27, df = 2, *p* = 0.52). Figure 2A indicates that although fewer cases were reported in rural locations, these tended to have a higher mortality rate.

The main sources of plastics reported were netting, bags, fencing, and rings from bottles. Survivorship varied between plastic types (χ^2^ = 14.47, df = 5, *p* = 0.01). Although plastic netting was the most frequently recorded type of plastic (*n* = 114/184) (Figure 2B), most individuals entangled in this way recovered (69/114). The highest mortality rates were associated with bands (hair bands and elastic bands) and yoghurt pots. The category ‘Other’ included a variety of sources that were generally associated with high fatality rates, such as plastic fencing (2/5), bailer twine (2/2), and cables ties (2/5).

### Mortality Model

Using the population dynamics model, key parameter estimates from the literature, and the survey of 52 rescue centres, we estimate that plastic entanglement in hedgehogs is responsible for between 1400 and 7999 hedgehog deaths annually in Great Britain (Table 2). This is an additional 1.4% compared with baseline estimated mortality.

The mortality can be attributed to three causes in proportions: $D_{i}/D_{T}$ for cause ‘i’. allowing us to predict the population dynamics (e.g., growth rate) if one cause was removed.

If the population suffers all causes of deaths, then $\lambda_{all\;causes}$ is found as the dominant eigenvector when $\mu_{2}$ takes the values found in Table 2. As expected, this leads us to retrieve an exponential growth rate from −8% to 0%. If the population does not suffer from RTAs, then $\lambda_{noRTA}$ is found as the dominant eigenvector when $\mu_{s}$ takes the values $\mu_{s}^{noRTA}=\mu_{s}\frac{D_{R}+D_{O}}{D_{T}}$. Similarly, we can obtain $\lambda_{noP}$ and $\lambda_{noRTA\_P}$, or the growth rates when no plastic entanglement occurs, or no RTAs nor plastic entanglement occur. Assuming the baseline mid-points for f and $D_{R}$ (i.e., f=4.5 and $D_{R}=74,000$), and two scenarios of high and low growth rate (solid vs. dashed lines in the figure for low and high growth rate), we can predict the growth rate, population structure, and population dynamics for each of the four mortality scenarios (baseline with all causes of mortality, and three counterfactual scenarios).

The predicted dynamics (Figure 3) indicate that if plastic entanglement deaths are removed, the wild population of hedgehogs is likely to increase slightly, although this increase is much smaller than would be observed with the cessation of road traffic accidents.

## 4. Discussion

This study demonstrates that plastic entanglement accounts for 1.4% of hedgehog admissions to British wildlife rehabilitation centres annually, and 46% of these animals die. Responses were received from one-third of the centres contacted—a good response rate for a questionnaire survey—and even had the non-responding centres recorded zero cases, appreciable numbers of hedgehogs would still be affected annually. The true rate of entanglement will be higher than that reported here, since members of the public may assist without contacting rescue centres, and entangled hedgehogs are less likely to be found than road traffic casualties, which have been widely studied [45,46,47]. An estimated 879,000 hedgehogs are present in Britain [42], and our models indicate that plastic entanglement is likely to have a negative overall impact on the population. While the scale of the impact is much lower than that attributable to road traffic accidents, it is nevertheless an additional pressure on a species that is considered vulnerable to extinction on the GB Mammal Red List [35].

Plastic netting often used in gardens, agriculture, and allotments was the most common cause of plastic entanglement, which is unsurprising as hedgehogs are likely to spend significant time in these locations, and the netting is deployed low to the ground. As this type of plastic is so often used by gardeners and farmers, this could suggest hedgehogs entangled in this plastic are more frequently found, and therefore have a better survival rate compared to those entangled in other plastics. Studies of marine mammals have also identified netting as one of the most common causes of entanglement and one that is linked with higher mortality rates [12,22,27,48]. Plastic bags and plastic rings from can holders were found to cause most of the remaining entanglement cases, which is also comparable with evidence of high rates of marine mammal entanglement in single-use plastics [17,48]. The plastics with the highest mortality rates, however, were bands, including elastic bands and hair bands. These cases had an 85% fatality rate. This is comparable with the serious and often life-threatening injuries reported in sea lions and seals because of entanglement with bands [49,50]. It is possible that elastic bands are particularly problematic for two reasons: first, it is difficult for wild animals to extricate themselves once the band is in place; and second because the bands cause damage to tissues and nerves and can also constrict the blood supply. Recent studies of pinnipeds have indicated that bands are often the most common type of plastic for them to become entangled in [12,48]. These studies have also found that juveniles are five times more likely to become entangled, whereas pups and adults are the least likely [12,48]. Other studies investigating species including crows, turtles, blue sharks and Antarctic fur seals also reported that juveniles are the most common age class to be entangled in plastic [29,51,52,53].

Entangled hedgehogs were rescued less frequently in rural than urban areas, potentially reflecting lower hedgehog population densities in rural areas and also the lower probability of being found owing to the lower human population. This is similar to many other studies that also report higher admission rates from urban than rural locations [54,55,56]. The prognosis was slightly poorer for animals in rural areas, possibly because there is likely to be, on average, a longer interval between entanglement and being found. The precise geographical location of the casualty was unknown for five individuals, so the habitat type with a 15 km radius of the centre (the maximum distance from which casualties were accepted) was used as a proxy. Given the small numbers of individuals involved, it is unlikely that this would materially affect the results.

This study is the first to indicate that plastic entanglement is causing serious welfare issues for the European hedgehog and results in high mortality rates. Therefore, we suggest that a national database is established to enable rescue centres and members of the public to record all incidents of plastic entanglement, allowing for future assessments to be made on a wider geographical scale. The database could also collate information on the sex and age profiles of casualties, together with more detailed information on the type of plastic involved, to enable more comprehensive assessments to be made of the risks of plastic entanglement.

## 5. Conclusions

This study indicates that, although understudied within terrestrial environments, plastic entanglement poses a welfare issue for an estimated 1400–7999 hedgehogs in Great Britain alone, and poses a conservation threat to populations already at risk. The development and use of the national database would facilitate better understanding of the true rates of plastic entanglement in wild populations.

## Figures and Tables

**Figure 1 animals-13-02448-f001:**
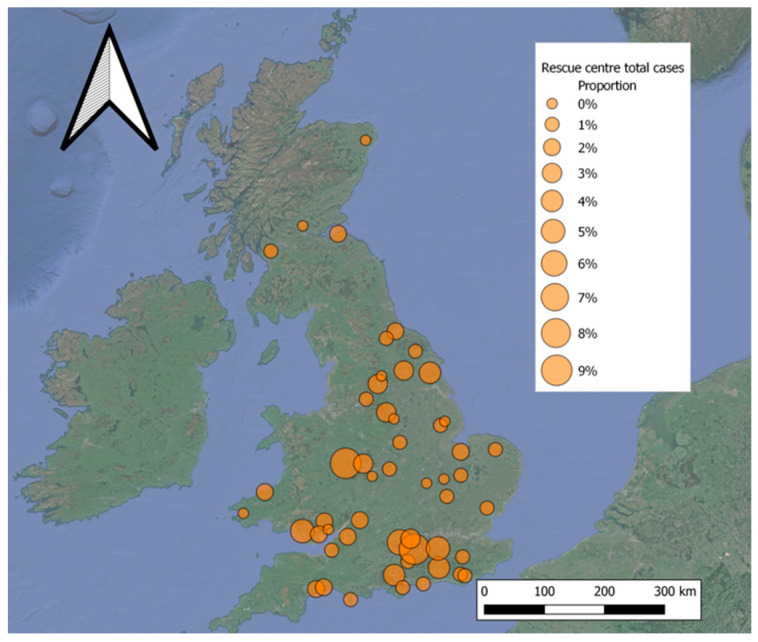
Map showing the proportion of admissions recording entanglement for each of the 52 rescue centres.

**Figure 2 animals-13-02448-f002:**
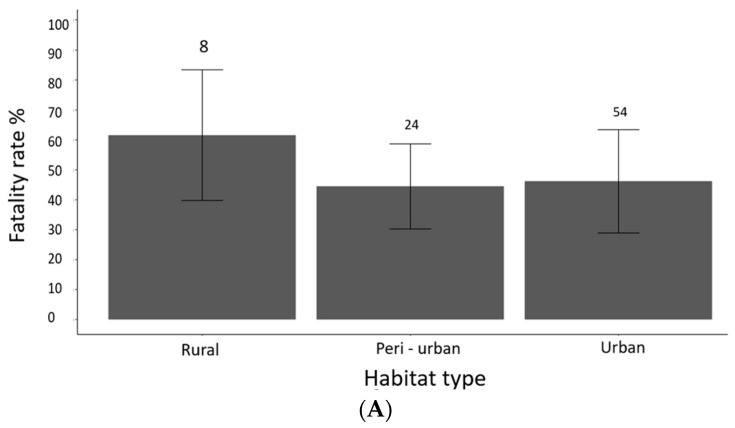
Plot showing the fatality rate from plastic entanglement, (**A**) by habitat type, (**B**) by plastic type. Error bars show 95% Wilson Confidence Intervals.

**Figure 3 animals-13-02448-f003:**
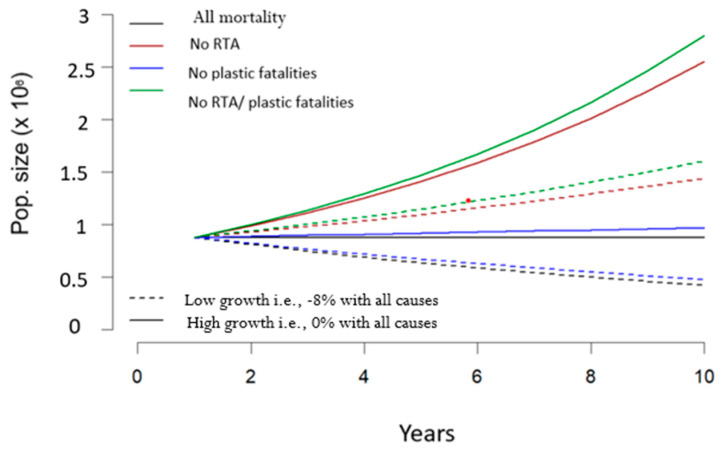
Predicted dynamics (population size over the 10 years) when growth rate is low or high., i.e., leading to a [0; −8%] growth rate when all mortality causes are accounted for (solid vs. dashed lines). The predicted dynamics are shown when all mortality is present (black), as well as 3 counterfactual scenarios, e.g., (1) if no RTAs occur (red), (2) if no plastic entanglement occurs (blue), (3) if no RTAs nor plastic entanglement fatalities occur (green).

**Table 1 animals-13-02448-t001:** Parameters used for the model.

Parameter		Reference
Sexual maturity	From 2 years old	[37]
Average litter size—*f*	4–5	[38,39]
Mortality to 15 weeks *μ*_0_	0.5	[38,39]
Mortality to 1 year old—*μ*_1_	0.5	[39,40]
Exponential growth rate,	−8% to 0%	[41]
Mortality for individuals >1 year old, *μ*_2_ = *μ*_3_	To be estimated	
Total population size, *N*	879,000	[42]
Deaths linked to Road Traffic Accidents *D_R_*	1.1% Low	[37,43,44]
	8.8% Medium	
	55% high	

**Table 2 animals-13-02448-t002:** Estimated adult mortality in Great Britain and cause of deaths in hedgehogs based on assumptions. *f* = average litter size, *μ*_0_ = mortality to 15 weeks, *μ*_1_ = mortality to 1 year old, log(*λ*) = Exponential growth, *N* = Total population size, *D_R_ =* Deaths linked to road traffic accidents, *μ*_2_ = *μ*_3_ = mortality for individuals >1 year old, *D_T_* = Deaths associated with all causes, *D_p_* = deaths associated with plastic and *D_o_* deaths associated with other causes.

Assumed						Estimated			
*f*	*μ* _0_	*μ* _1_	log(*λ*)	*N*	*D_R_*	*μ*_2_ = *μ* _3_	*D_T_*	*D_O_*	*D_P_*
4	0.5	0.5	−0.08	879,000	9.67 × 10^3^	0.401221	599,448.5	581,787.9	7991.592
					7.73 × 10^4^	0.401221	599,448.5	515,022	7074.478
					4.38 × 10^5^	0.401221	599,448.5	114,426.7	1571.795
			0	879,000	9.67 × 10^3^	0.333322	597,719.7	580,082.6	7968.167
					7.73 × 10^4^	0.333322	597,719.7	513,316.7	7051.053
					4.38 × 10^5^	0.333322	597,719.7	112,721.4	1548.37
4.5	0.5	0.5	−0.08	879,000	9.67 × 10^3^	0.426385	594,704.7	577,108.4	7927.313
					7.73 × 10^4^	0.426385	594,704.7	510,342.5	7010.199
					4.38 × 10^5^	0.426385	594,704.7	109,747.2	1507.516
			0	879,000	9.67 × 10^3^	0.359991	592,813.7	575,243	7901.69
					7.73 × 10^4^	0.359991	592,813.7	508,477.1	6984.576
					4.38 × 10^5^	0.359991	592,813.7	107,881.8	1481.893
5	0.5	0.5	−0.08	879,000	9.67 × 10^3^	0.449562	590,586.8	573,046.3	7871.515
					7.73 × 10^4^	0.449562	590,586.8	506,280.4	6954.402
					4.38 × 10^5^	0.449562	590,586.8	105,685.1	1451.719
			0	879,000	9.67 × 10^3^	0.384617	588,547.9	571,035	7843.887
					7.73 × 10^4^	0.384617	588,547.9	504,269.1	6926.773
					4.38 × 10^5^	0.384617	588,547.9	103,673.8	1424.09

## Data Availability

Available online: https://doi.org/10.6084/m9.figshare.23600328 (accessed on 19 June 2023).

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
