# Peer review of "Plastic Entanglement Poses a Potential Hazard to European Hedgehogs Erinaceus europaeus in Great Britain"

_animals, 2023, doi:10.3390/ani13152448_

Round 1

Reviewer 1 Report

This study aims to quantify the rate and severity of European hedgehog plastic entanglement and to estimate the likely effects at the population level. The topic of the work is original and fills a gap in this field. This is an important topic as there is growing concern in the UK about the impact of plastic entanglement on the European hedgehog (Erinaceus europaeus). Plastic poses risks to both populations and individuals. Research so far has been in birds, although entanglement also occurs in mammals. Appropriate research methods were used in the work and the obtained data were subjected to proper statistical analysis. The conclusions are consistent with the obtained research results. The cited literature concerns the subject of the research. In my opinion, the work is valuable and can be accepted for publication in the journal Animals.

Reviewer 2 Report

Thank you very much for the opportunity to review the manuscript entitled “Plastic entanglement poses a potential hazard to European hedgehogs Erinaceus europaeus in Great Britain”. I found the topic of the manuscript very interesting and important for animal welfare. On the one hand, the topic is known, but on the other hand, there is a lack of research on that. We know that the mortality on roads can cause a decrease in the population reaching 50%, but based on the manuscript we should think is plastic the second (after road traffic) danger for the hedgehog population.

Detailed comments:

l. 23 I think there is a double space before “Population modelling…”

l. 28 Keywords should be changed because all of them are already used in the title. Based on the abstract I suggest changing keywords to: wildlife, rehabilitation centres, questionnaire, population modelling, plastic waste. These are only suggestions.

l. 42 Full stop is missing

l. 44 understanding of what?

l. 86 In the “Materials and methods” I couldn’t find the number of responses to the sent questionnaires, but in this line the number of cases without specific location is added. Based on that it’s hard to say what share of the whole data is that. Because in this chapter the materials should be included I suggest adding the number of rehabilitation centres that responded. In the Results, the Authors will give the number of cases (I understand that one hedgehog is one case), without rehabilitation centres. Moreover, I wonder if using the location of rehabilitation centres is correct. Are there data showing how far from the rehabilitation centre the animal could be found?

l. 101 I would write 1:1 ratio (I’ve never seen 1-1 description).

l. 151-159 It’s hard to follow all the percentages and values added in brackets, especially when they are divided by unknown values. I suggest skipping most of the percentages and just showing the raw data. Most of the percentages are not needed.

“Of the 160 rescue centres contacted, responses were received from 54 (Figure 1). These centres provided data on 12,546 hedgehog admissions. Ten centres reported zero admissions of plastic-entangled hedgehogs, and all of them reported annual admissions of fewer than 200 individuals. , however, there were also centres of similar annual admission sizes which reported entanglement cases.

Data from the 44 centres with entanglement cases showed  184 (out of 11,426)  admissions were a consequence of plastic entanglement. The survivorship (and re-release rate) was 53.2% for all 184 admissions.” Same for lines 176-182. If the Authors want to remain the value 1.4% (which seems to be the most important result, I suggest changing the text – for example: Data from the 44 centres with entanglement cases showed 184 admissions were a consequence of plastic entanglement, what gives 1.4% of all reported admissions.I suggest to skip the crossed sentence

Figure 2A. Please add the number of cases to names at the X-axis – Rural (n=8), Peri-urban (n=24), Urban (n=54). Same for Figure 2B. for per-urban location in the text the value of 16% is given, while on the graph is around 40%. There is no need to give the all numbers in the text (in that case the value is not necessary). Moreover, I would give the raw value to the graph 2B. For three categories the total number of cases is less than ten. I suggest giving bars with the total cases for all categories, with different colours for cases of mortality (on the same bars). It would show the fatality rate but also the real numbers of cases for all categories. Why “other” is in the middle of the graph? Please change the order of categories according to real value (if the Authors change the type of graph). The “Other” category should be always at the end.

See the example: (https://stackoverflow.com/questions/57505493/how-to-combine-2-variables-in-bar-chart-by-using-ggplot-in-r-studio)
Also shown in the attached file.

l. 173 Please ad a full stop after “by plastic type”

l. 188 “This is an additional 1.4% compared with baseline estimated mortality.” I don’t understand the sentence. What is “baseline estimated mortality”.

Table 2. All tables and graphs should have all the necessary information to explain them, please add all the shortcuts from the table under it.

l. 243 After reading the whole manuscript, I can’t tell where those values were shown in the Results.

It is very important to show all the raw values in the Appendix. For example as a table

rescue center

Number of hedgehog admissions

No. of admissions of plastic-entangled hedgehogs

Number of survivorship (and re-release)

1

54

If annual admission is another number (based on the Manuscript I assume it is), please add it to the table.

Overall, I think that the Manuscript is very interesting, and after some revision. I wish the Authors all the best.

Reviewer 3 Report

This is a nice attempt to quantify and assess a quite well-known problem and I can see ways it can be useful for dealing with the problem of plastics in terrestrial habitats beyond Great Britain and beyond hedgehogs. I like the way rehabilitation centre data and a demographic analysis can be used to produce useful guidelines for future conservation efforts.

Overall I am quite happy with the content of the manuscript as it is. I just note a few points with typos or presentation issues that might improve the final paper:

1. Correct ‘morality’ to ‘mortality’ in the following three instances:

Page 2, Line 90: 2.4. Morality model”

Page 8, Figure 4, legend (1st curve): “All morality”

Page 8, Line 213 (Figure 4, subtitle): “when all morality is present”

2. Figure 2: The labels above the error bars and the labels of the two axes are far too small to read. Increase font size.

3. Figure 4: I would consider an alternative labelling of the y-axis, with axis title changed to something like “Pop. size (x 106)” and labels changed to 0, .5, 1.5, 2.5.

Reviewer 4 Report

The authors present an interesting (although brief) study on the true impact of plastic residues on hedgehogs in Great Britain by analysing questionnaires from rescue centres. Even though the topic raises attention nowadays and it is brand new information in hedgehogs biology and conservation, I believe the authors do not provide a deeper analysis of the problem for this population. In other words, many questions remain unanswered without proposing further research in DIscussion or conclusions. Therefore, I think authors should consider turning their paper into a Short Communication or, alternatively, provide a better and more detailed interpretation of their results. In detail:

1) Introduction should also provide a background on the toxicity of some chemical hazards commonly associated with plastic that may cause acute or chronic signs of toxicity (as additives and preservatives), because this is also a dangerous aspect of plastic, besides the physical effect of them when absorbed by animals. 

2) The Discussion section is poor from my perspective. The authors present a lot of data in the Results section but this data is not discussed in detail or properly compared with other works. It would be also interesting to discuss which plastic is the most dangerous (type, morphology, etc...), if females or males, hoglets or adults are more affected by this pollution and why. How was your questionnaire designed? How many questions does it have? What did you ask our colleagues regarding this problem? Only mortality rates? If you do not have these data you can at least suggest this for the future or compare it with other studies in other species and rise the hypothesis. 

Also, the authors should also discuss their study limitation regarding not knowing the exact provenance of some hedgehogs and using the location of the rescue centres because this can influence the interpretation of the next results. 

Once again, I honestly believe the study and the topic have a remarkable importance to be addressed especially because you found out that the impact of plastic on these populations is considerable.  However, I also believe some more details can be provided or at least discussed properly.

Round 2

Reviewer 4 Report

The authors have addressed all my comments. I have nothing else to add.